# Cerebral Cortex Activation and Gait Performance between Healthy and Prefrail Older Adults during Cognitive and Walking Tasks

**DOI:** 10.3390/brainsci13071018

**Published:** 2023-06-30

**Authors:** Weichao Fan, Chongwu Xiao, Longlong He, Ling Chen, Hang Qu, Qiuru Yao, Gege Li, Jinjing Hu, Jihua Zou, Qing Zeng, Guozhi Huang

**Affiliations:** 1Department of Rehabilitation Medicine, Zhujiang Hospital, Southern Medical University, Guangzhou 510280, China; fanfan0824@smu.edu.cn (W.F.); 19811941276@163.com (C.X.); 957995818@smu.edu.cn (L.H.); chenling202204@163.com (L.C.); 15265420979@163.com (H.Q.); y3697707@163.com (Q.Y.); lggisme@163.com (G.L.); h19974198131@smu.edu.cn (J.H.); 2School of Nursing, Southern Medical University, Guangzhou 510280, China; 3School of Rehabilitation Medicine, Southern Medical University, Guangzhou 510280, China; 4Faculty of Health and Social Sciences, The Hong Kong Polytechnic University, Hong Kong 999077, China

**Keywords:** aging, pre-frailty, executive functions, dual-task, gait, fNIRS

## Abstract

Pre-frailty is a transitional stage between health and frailty. Previous studies have demonstrated that individuals with pre-frailty experience declines in cognitive and gait performances compared with healthy individuals. However, the basic neural mechanism underlying this needs to be clarified. In this cross-sectional study, twenty-one healthy older adults and fifteen with pre-frailty underwent three conditions, including a single cognitive task (SC), single walking task (SW), and dual-task (DT), while cortical hemodynamic reactions were measured using functional near-infrared spectroscopy (fNIRS). The prefrail group (PG) showed a significantly lower activation of the left dorsolateral prefrontal cortex (L-DLPFC) than the healthy group (HG) when performing SC (*p* < 0.05). The PG showed a significantly lower Timed Up and Go test and step speed than the HG during SW (*p* < 0.05). The coefficient of variation (CV) of the step length of the PG was significantly higher than that of the HG when performing DT (*p* < 0.05). No significant correlation in cerebral cortex activation and gait parameters in the HG when performing SW and DT was noted (*p* > 0.05). Participants of the PG with a higher oxygenated area in the left anterior prefrontal cortex (L-APFC) had a lower step frequency during SW (*r* = −0.533, *p* = 0.041), and so did the following indicators of the PG during DT: L-APFC and step speed (*r* = −0.557, *p* = 0.031); right anterior prefrontal cortex and step speed (*r* = −0.610, *p* = 0.016); left motor cortex and step speed (*r* = −0.674, *p* = 0.006); step frequency (*r* = −0.656, *p* = 0.008); and step length (*r* = −0.535, *p* = 0.040). The negative correlations between the cerebral cortex and gait parameters of the PG indicated a neural compensatory effect of pre-frailty. Therefore, older adults with pre-frailty promote prefrontal activation to compensate for the impaired sensorimotor systems.

## 1. Introduction

Frailty is a multidimensional and significant geriatric syndrome that is theoretically defined as a clinically recognizable state of increased vulnerability, which is caused by the decline in reserves and functions of multiple physiological systems related to aging [1]. The overall prevalence of frailty among older adults in Chinese communities is 10% [2]. Moreover, frailty may increase the risk of mild cognitive impairment and Alzheimer’s disease and is essential in the context of global aging [3]. Frailty is a dynamic process. Pre-frailty is a transitional stage between health and frailty in clinical practice, and a recent study has shown that the prevalence of pre-frailty was 46% for physical frailty [4,5]. It is necessary to screen and identify the indicators and neural mechanisms of pre-frailty, which is key to preventing and intervening in frailty [6,7,8].

Cognition and gait are significant components of pre-frailty assessment. Cognition incorporates multiple dimensions, including executive function. Working memory is one of the cores of executive function and is inherently involved in all higher-level cognitive activities [9]. A previous study reported that individuals with pre-frailty presented significantly higher risks of impairments in working memory than robust persons. More evidence is needed to further confirm the impaired mechanism in prefrail older adults [10]. Meanwhile, slow gait, which is one of the frailty elements, can precede cognitive decline [10]. Gait disturbances can reflect dysfunction in the nervous system and are closely related to frailty. Gait has been recommended as a marker of central and peripheral functions in several geriatric studies. Studying various gait parameters may enhance frailty classifications and identify motor function impairments in frailty [11].

Cognitive and walking dual-task (DT) performance is an effective method for the accurate assessment of cognitive and gait performances. It refers to the simultaneous execution of a cognitive task during walking, which is close to the actual scenario of daily life activities. Cognitive and walking DT performance may indicate a marker of cognitive and motor function decline [12]. A recent study demonstrated that DT exerted difficulty on pre-frailty. However, the study lacked a comparison with healthy older adults and failed to measure cognitive performance during walking tasks [13]. Therefore, more evidence is needed to show the difference in cognitive and gait performance during DT between healthy and prefrail older adults.

Previous researchers have used traditional neuroimaging techniques (e.g., magnetic resonance imaging) to study the relationship between brain function and pre-frailty. The prefrail state is associated with global brain atrophy and white matter hypointensities [14,15]. In older adults, the global and regional cortical thickness was cross-sectionally associated with frailty, and those with higher global cortical thickness were less likely to be prefrail [16]. However, owing to the problems of traditional neuroimaging methods in some paradigms (e.g., walking), studies on related neural activities in pre-frailty remain challenging. Recently, functional near-infrared spectroscopy (fNIRS) research technology has circumvented this limitation. fNIRS is a portable brain imaging technique that enables noninvasive long-term measurements of cortical hemodynamic responses (e.g., changes in oxygenated and deoxygenated hemoglobin levels) in natural situations, even during exercise [17,18]. Since fNIRS is relatively motion-tolerant and environmentally unconstrained, it is a promising tool for fostering the development of aging biomarkers and discovering neural mechanisms [19].

The integrity of the cerebral cortical areas is critical for preserving cognition and mobility late in life. Older adults show increased neural activity to meet behavioral needs [20]. Moreover, previous studies have shown that frail older adults exhibit cortical atrophy, possibly due to their neurological inefficiency [16]. Cortical gait control during aging is bilateral, widespread, and dependent on cortical integrity. Considering existing theories of cognition and brain reserve, the frontal cortex activation in individuals with gait disorders is evident when faced with increased demands for walking and cognitive tasks, which may be related to neural compensation effects [21]. Therefore, it is essential to understand the central mechanisms of cognitive and gait deficits associated with health and pre-frailty. Understanding this difference may provide necessary help for the early recognition of pre-frailty. 

In light of the above, we aimed to explore differences in cerebral cortex activation, gait parameters, and working memory performance between healthy and prefrail older adults when performing cognitive and walking tasks. Furthermore, we aimed to explore the association between cerebral cortex activation and gait performance. We hypothesized the following: (a) Differences in prefrontal cortex and motor cortex activation between healthy and prefrail older adults when performing tasks would exist. The gait and working memory performance of healthy older adults would be better than those of prefrail older adults. (b) The prefrontal cortex and motor cortex activation would be correlated with gait parameters in each group. These findings may provide neuroimage and clinical clues for early detection and help discover the underlying neural mechanism of cognitive and gait decline in pre-frailty.

## 2. Material and Methods

### 2.1. Participants

This was a cross-sectional study. We used a moderate effect size of 0.4 (f), a power of 0.80, an ɑ-level of 0.05, and a correlation among the repeated measures of 0.4. The power analysis indicated that a minimum of 30 participants was required (at least 15 per group). In our study, a total of 36 participants, including 21 healthy older adults in the healthy group (HG) and 15 with pre-frailty in the prefrail group (PG), were recruited from the community. The following were the inclusion criteria: (a) age ≥ 60 years; (b) right-handed; and (c) the appearance of one or two of the five physical characteristics shown in Table 1 defined as pre-frailty based on Fried et al.’s frailty phenotype criteria [22]. The following were the exclusion criteria: (a) abnormal vision and hearing; (b) limb movement disorder and obstacles to walking; and (c) stroke, diabetes, peripheral neuropathy, orthopedic disease, malignant tumor, brain disease, acute cerebrovascular disease, mental disorders, and cognitive impairment (Montreal Cognitive Assessment [MoCA] score < 24 points). All participants signed the informed consent form. The study protocol was approved by the Medical Ethics Committee of Zhujiang Hospital, Southern Medical University, Guangzhou, Guangdong, China, (2022-KY-083-01).

### 2.2. Clinical Measurements

Participants’ cognitive status was measured using the MoCA scales [23]. The use of the MoCA scales in our study was officially licensed. Depression symptoms of participants were screened using CESD-10, which has shown satisfactory reliability and validity in older Chinese adults [24].

The Timed Up and Go (TUG) test was used to measure the physical function of participants and was measured in seconds. The test required an individual to stand up from a standard armchair (the seat height was approximately 46 cm), walk 3 m away, turn around, walk back to the chair, and sit again. Participants wore regular shoes, and no physical assistance was provided. They started by positioning their back against the chair with their arms at the arms of the chair. When they were asked to “go”, they stood up and walked comfortably and safely to a line on the floor 3 m away, then turned around, returned to the chair, and sat again [25]. For familiarization, the participants performed the test once before being timed. The test was performed three times, and the average value was taken.

IPAQ-SF was used to evaluate the physical activity level of participants. The questionnaire contains seven items, which are used to calculate the metabolic equivalent to assess the physical activity level, and collects the time spent in sedentary behavior. The Chinese version of IPAQ-SF has been verified in previous studies and has good reliability and validity [26].

Participants’ hand grip strength (HGS) was measured using a grip strength meter (CAMRY EH101, Senssun Weighing Apparatus Group Ltd., Guangdong, China). The participants were seated in a comfortable chair without arm support, with the elbow in 90° flexion, and the upper arm and lateral thorax were separated to ensure accuracy [27]. They were subsequently instructed to hold the instrument with their left and right hands as hard as possible once, and the maximum value was used for analysis [28].

### 2.3. The 2-Back Task

The n-back paradigm is widely used in working memory research. When performing n-back tasks, participants need to decide whether the presented items (e.g., number) are the same or different from the previous *n* trials in each trial. Generally, *n* varies between 1 and 3. We selected the 2-back task, and a study has shown that it provides older adults with the best level of cognitive challenges [29].

In our study, the numbers (1–9) were played on the computer using E-Prime 3.0 (Psychology Software Tools Inc., Sharpsburg, PA, USA). It presented 20 numbers in a pseudo-random order, with each stimulus lasting 0.5 s and 1 s apart. Participants were instructed to listen to the string of numbers; when they heard the same number as the stimulus two positions prior, they pressed the “Yes” button, and the “No” button if otherwise (the button was held in their right hand). Before the experiment started, the participants briefly practiced the 2-back task until they could correctly respond to at least 70% of the single cognitive (SC) task stimulus [30]. We calculated the accuracy and reaction times of the 2-back task as indicators of the participants’ working memory performance. Accuracy and reaction times can be used to measure performance on the task [31].

### 2.4. fNIRS Measurements and Data Processing

A portable 35-channel near-infrared functional brain imaging device (NirSmart, Danyang Huichuang Medical Equipment Co., Ltd., Danyang, China) was used to record cerebral cortex activation. This device has been effectively used in previous studies [32]. Moreover, the manufacturer provides more relevant information online (http://hcmedx.cn/en/), accessed on 5 May 2023. The system consists of a near-infrared light source (light-emitting diodes) and avalanche photodiodes as detectors, with 730 and 850 nm wavelengths, respectively, and an 11 Hz sampling rate. The experiment uses 13 sources and 15 detectors to form 35 channels, and the average distance between the source and the detector is 2.7 cm, with reference to the international 10/20 system for positioning. The acquired coordinates were then transformed into Montreal Neurological Institute coordinates and further projected to the Montreal Neurological Institute standard brain template using a spatial registration approach in NirSpace (Danyang Huichuang Medical Equipment Co., Ltd., Danyang, China). The mean root square static accuracy of the positioning system was 0.06 inches and the orientation was 0.40 degrees. The resolution was 0.00046 inches, the orientation was 0.0038 degrees, and the effective range was 36 inches. The participants carried a unique backpack to load the fNIRS host machine, which transmits the data to the computer through wireless mode, as shown in Figure 1A,B.

As described in previous studies, we measured the participants’ left anterior prefrontal cortex (L-APFC), right anterior prefrontal cortex (R-APFC), left dorsolateral prefrontal cortex (L-DLPFC), right dorsolateral prefrontal cortex (R-DLPFC), left motor cortex (L-MC), and right motor cortex (R-MC) as regions of interest since these are related to cognition and motion functions [20,33]. Moreover, we focused on the oxyhemoglobin (HbO_2_) concentration as a marker of cortical activity because it is the most sensitive and reliable indicator of locomotion-related changes in regional cerebral oxygenation [34,35].

The NirSpark software package (Danyang Huichuang Medical Equipment Co., Ltd., Danyang, China) was used for analyzing fNIRS data, as described previously. First, the raw optical intensity data were converted to optical density data. Second, spline interpolation was used to remove motion artifacts [36]. Motion artifacts were manifested as impulse or cliff-type jumps caused by the relative sliding of the scalp and probes. Then, the raw data were band-pass-filtered between 0.01 and 0.2 Hz to remove physiological noise (e.g., respiration, cardiac activity, and low-frequency signal drift). Subsequently, the denoised optical density data were converted into hemoglobin concentration data. Finally, the modified Beer–Lambert law was used to calculate the relative hemoglobin concentration changes in oxygen–hemoglobin [37]. The HbO_2_ concentrations for each block paradigm were superimposed and averaged to generate a block average result.

### 2.5. Gait Measurements

The gait parameters, including step speed, step frequency, and step length, were measured using the Intelligent Device for Energy Expenditure and Activity (IDEEA^®^, MiniSun LLC, Fresno, CA, USA) equipment. This device used five advanced three-dimensional motion sensors (acceleration sensor) to ensure high-precision recording. Moreover, the main recorder of the IDEEA system was secured on the left waistband, one sub-recorder was taped above each lateral malleolus, and five sensors were placed on the sternum and bilaterally on the plantar aspect of the foot and midline of the anterior aspect of the thigh [38]. The IDEEA system is a convenient wearable sensor with excellent reliability and has been verified in the study of gait performance measurement among older adults [39].

We calculated gait variability using the coefficient of variation (CV), which is the ratio of the standard deviation to the mean multiplied by 100 (CV = [{standard deviation/mean} × 100]). Gait variability quantifies the gait automaticity, with more significant variability usually indicating the loss of gait regulation, causing irregular and unstable gait [40].

### 2.6. Procedures

The dual-task protocol consisted of the following three conditions: SC task, single walking task (SW), and dual-task (DT). The SC required the participants to perform the 2-back task while standing. During the SW, the participants were asked to wear their own shoes and walk comfortably and straightly in a long corridor at their usual speed. The corridor was a cement floor with normal lighting and a length of 40 m. During the DT, the participants were asked to complete the 2-back task while walking. Moreover, they were instructed to pay equal attention to both tasks to minimize task prioritization effects [41]. fNIRS recording was conducted at baseline while resting in the initial position for 30 s. Then, each experimental condition was managed in a block of 30 s; subsequently, the participants were instructed to rest for 30 s. We repeated each task four times, and the complete sequence of the stimulus block was presented according to the A-B-B-A design. This type of design can effectively control the fatigue effect of the participants under various conditions; the reliability and validity of this paradigm have been well-established [42,43]. The same computer automatically and synchronously collected the data of fNIRS, gait, and 2-back task. The experimental procedures are presented in Figure 1C.

### 2.7. Statistical Analysis

The normality (Shapiro–Wilk test) and homogeneity of variance (Levene test) of data were tested to observe if parametric analysis assumptions were met. Data were presented as *n* (%) and mean (standard deviation) for categorical and normally distributed variables, respectively. When normality was reached, one-way analysis of variance (ANOVA) was used to test the significant difference in cerebral cortex activation, gait parameters, and 2-back task performance between the two groups. Otherwise, a non-parametric test was considered. *p* values were applied using the Bonferroni multiple comparison correction. Categorical variables were compared using the chi-squared test. The Pearson correlation analysis method was used to analyze the correlation between HbO_2_ and gait parameters. Correlations were adjusted for multiple comparisons. All analyses were performed using the Statistical Package for the Social Sciences (version 23, IBM, Armonk, NY, USA). *p* < 0.05 was considered statistically significant.

## 3. Results

### 3.1. Participants’ Characteristics

A total of 36 participants were enrolled in this study. The average age of the HG and PG was 65.95 and 67.93 years, respectively. The PG had a significantly longer TUG test than the HG (*p* < 0.05). Gender, age, body weight, body height, BMI, years of education, HGS, MoCA, CESD-10, and IPAQ-SF were not significantly different between the groups (*p* > 0.05) (Table 2).

### 3.2. Cerebral Cortex Activation

One-way ANOVA showed that when performing SC, the HbO_2_ concentration of L-DLPFC of the HG was significantly higher than that of the PG (*p* < 0.05), and no significant difference was observed in the activation of other regions (*p* > 0.05). No significant difference in cerebral cortex activation was noted between the HG and PG when the participants performed SW and DT (*p* > 0.05) (Figure 2). The cerebral cortex activation of a typical participant in each of the two groups when performing the SC, SW, and DT is presented in Figure 3.

### 3.3. The 2-Back Task Performance

No significant difference was noted between the HG and PG in the accuracy and reaction time of performing SC and DT (*p* > 0.05) (Table 3).

### 3.4. Gait Performance

The HG had a significantly higher step speed than the PG when performing the SW (*p* < 0.05). The CV of the step length in the PG was significantly higher than that in the HG when performing DT (*p* < 0.05). No statistical difference was observed in other gait parameters (*p* > 0.05) (Table 4).

### 3.5. Association between Measured Cerebral Cortex Oxygenation and Gait Parameters

Correlation analysis showed that no significant correlation in cerebral cortex activation and gait parameters was observed in the HG when performing SW and DT (*p* > 0.05) (Figure 4A).

Correlation analysis showed that the L-APFC and step frequency of the PG were moderately negatively correlated during SW (*r* = −0.53, *p* = 0.041). When performing DT, the following indicators of the PG were moderately negatively correlated: L-APFC and step speed (*r* = −0.56, *p* = 0.031); R-APFC and step speed (*r* = −0.59, *p* = 0.016); L-MC and step speed (*r* = −0.67, *p* = 0.006); step frequency (*r* = −0.66, *p* = 0.008); and step length (*r* = −0.53, *p* = 0.040) (Figure 4B).

## 4. Discussion

Previous studies have reported that there existed a decline in cognitive and gait performances in prefrail older adults [10]. To determine the possible neural mechanism of this decline, we designed the current study to explore the differences in cerebral cortex activation, gait parameters, and working memory performance between the HG and PG during SC, SW, and DT. Additionally, we examined the association between the cerebral cortex and gait performance. We observed that the two groups differed in cerebral cortical activation during SC. The PG performed worse in some gait parameters during SW and DT. More interestingly, we noted a correlation between the cerebral cortex activation and gait parameters in the PG although not in the HG. These findings are discussed in detail below.

The prefrail state, which is the early stage of physical frailty, is associated with brain structural changes in cognitively unimpaired older adults [14]. As the brain structure changes, the brain hemodynamics are altered, thereby causing changes in cerebral cortex region activation. Hypoactivation of the cerebral cortex in prefrail older adults may be a neural mechanism underlying their cognitive function decline. In this study, we noted that the L-DLPFC activation in the PG was significantly lower than that in the HG during SC. The likely cause of the PG showing L-DLPFC low activation is neurodegenerative disorders (e.g., cortical atrophy) [16,44]. Moreover, a model of hemispheric asymmetry reduction in older adults [45] can be used to explain this result. In this model, young adults exhibit a marked predominance of unilateralization (e.g., asymmetry) in cerebral cortex activation when manipulating cognitive tasks, whereas older adults exhibit a marked bilateralization (e.g., asymmetry reduction) in cerebral cortex activation. The age-related asymmetry reduction could help counteract the age-related neurocognitive decline. To counteract neurocognitive deficits, older adults recruited both hemispheres to perform a task that requires one hemisphere in younger adults [46]. In our study, the R-DLPFC of the two groups showed no significant difference; however, the *p*-value was close to 0.05. A previous study reported that the R-DLPFC activation in the PG was lower than in others [28]. Overall, we speculate that the whole DLPFC activation of the PG is lower than that of the HG when performing SC. Another study showed that some neurocognitive symptoms are associated with decreased activation in the bilateral PFC [47]. Exploring whether the whole DLPFC activation is lower in the PG required further research. Furthermore, we observed increasingly pronounced activation of the DLPFC with increasing task difficulty, which was consistent with the previous conclusion that DT requires more attentional resources and induces higher brain activities than single-task [48]. However, this is only an increasing trend in our study, and whether statistical differences exist needs further research. Regarding APFC activation, no significant difference was noted between the two groups during SC, SW, or DT. This result was inconsistent with our assumption. Previous studies have shown that APFC is mainly responsible for regulating emotional behavior [49], which may not be sensitive to cognitive motor DT.

We observed that the gait performance of the HG was better than that of the PG. The step speed in SW and the step length CV in DT showed a significant difference between the two groups. A previous study demonstrated that step speed could reflect the prefrail state of older adults [50], and step speed is the most sensitive parameter to discriminate pre-frailty [51]. Moreover, prefrail older adults have difficulty performing simple tasks [52], which is consistent with the poorer performance in the PG during SW. Apart from step speed, we also performed the TUG test. The results showed that the HG spent significantly less time than the PG. The result of the TUG test indirectly reflected the difference in step speed. A recent study reported that both the TUG and step speed tests could be used to identify frail older adults with good accuracy [53]. Using our results, we suggest that both the TUG and step speed tests can accurately identify prefrail older adults. A recent study showed that step length CV, indicating the fluctuation in step length from one gait cycle to the next, is considered a strong predictor of negative outcomes [54]. In our study, the step length CV of the HG was significantly smaller than that of the PG during DT; however, this difference was absent during SW, confirming the predictive role of gait CV and dual-task for poor performance in prefrail older adults, with the effect being more pronounced when both were used in combination. Furthermore, a characteristic of pre-frailty or frailty is the decrease in step length [55]. In our study, the step length of the PG during DT was shorter than that of the HG. Although no difference was noted, the *p*-value (*p* = 0.057) was very close to 0.05; therefore, we believe that there will be more interesting findings following sample size expansion. Gait is closely linked to the motor cortex. In our study, significant differences were noted in gait parameters although not in the motor cortex, indicating that gait parameters are more sensitive to be used as a predictor than motor cortex activation in the prefrail stage of older adults.

In the present study, no difference was observed in the working memory capacity (WMC) between the two groups. However, previous studies have shown impaired executive performance and low WMC in prefrail older adults. More specifically, prefrail older adults seem to have poorer performance than healthy older adults in terms of various cognitive tasks that assess working memory and other indicators, including global cognition, processing speed, executive function, and visuospatial function [56]. Performing n-back tests is a valid instrument to measure the WMC of older adults. In our study, we used the 2-back tests’ accuracy and reaction time to evaluate the WMC of the PG and HG. Although prefrail participants showed a slightly lower mean performance in accuracy and reaction time, we did not observe significant differences in the cognitive status of the two groups. This illustrates that WMC does not remarkably decline during the pre-frailty stage, and accuracy and reaction time may not be useful indicators of pre-frailty when performing 2-back tests, since they are less sensitive than cerebral cortex activation and gait parameters. Moreover, a previous study indicated that gait can precede cognitive decline and may be an early clinical marker of cognitive decline [10].

In several geriatric studies, since gait is easily observable, it has been recommended as a marker of cerebral cortex activation [57]. Furthermore, several studies have demonstrated the indispensable link between gait and cerebral cortex activation. Therefore, we analyzed the correlation between cerebral cortex activation and gait parameters. A significant moderate negative correlation was noted between gait and cerebral cortex activation, including the L-APFC, R-APFC, and L-MC, in the PG although non-existent in the HG. We suggested that the reason for this phenomenon was the more significant asymmetry reduction of cerebral cortex activation in prefrail older adults, which required more diffuse cerebral cortex regions to support the performance of the tasks. Owing to the decline in gait function performance, prefrail older adults needed to use more cerebral cortex activation to maintain a good gait performance during challenging cognitive and walking tasks. Moreover, this negative correlation was more present during DT rather than SW. These results are consistent with our expectation, since DT, which involves both locomotion and cognition, is bound to occupy more resources of the brain regions. In other words, the negative correlations between the cortical activity and gait parameters in the PG suggested a neural compensatory effect in pre-frailty. Prefrail older adults promote prefrontal activation to compensate for the impaired sensorimotor systems.

This study had some limitations. First, the age of the participants ranged from 60 to 70 years, and how older adults over the age of 70 performed remained unclear. Future research should recruit older adults in more age groups. Second, the total number of participants was only 36; therefore, selection bias may exist. Although the sample size was relatively small, we still obtained significantly different results. Further studies are needed to validate our findings and explore more possible indicators and neural mechanisms of impaired cognitive and motor functions in pre-frailty and frailty.

## 5. Conclusions

In this study, the fNIRS device was used to explore the underlying indicators and mechanisms of cognitive and gait impairments in pre-frailty during cognitive and walking tasks. The results showed that the L-DLPFC of the PG has a significantly lower activation during SC, and their TUG test, step speed, and step length CV are significantly poorer during SW and DT. Moreover, this study indicated that the APFC and L-MC activation is moderately negatively correlated with the step speed, step frequency, or step length during SW and DT in the PG. These findings are in addition to the knowledge of cerebral cortex activities, gait performance, and their relationships in prefrail older adults based on cognitive and walking tasks. Additionally, this study helps to understand the possible neural compensatory mechanism of the cognitive and motor function decline in pre-frailty. Lastly, our study showed the value of fNIRS as an evaluation tool for future research to explore more potential predictors and neural mechanisms in pre-frailty.

## Figures and Tables

**Figure 1 brainsci-13-01018-f001:**
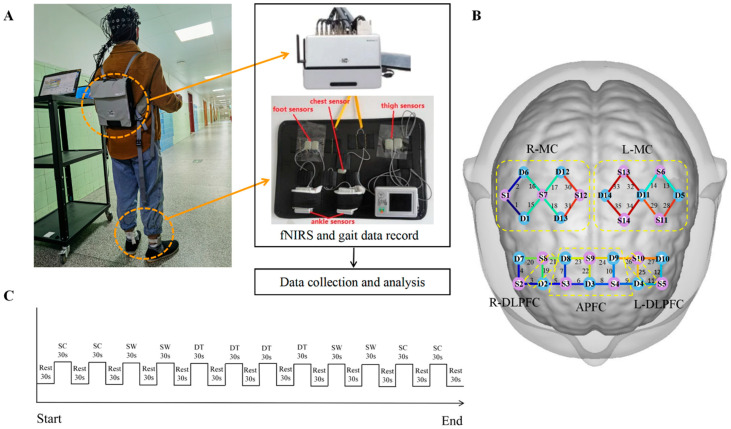
Experimental design. (**A**) The fNIRS and Intelligent Device for Energy Expenditure and Activity gait recording setup. (**B**) Probe configuration. The integers on the cerebral cortex indicate the recording channels (CHs). The purple dots are the sources, and the blue dots are the detectors. APFC: anterior prefrontal cortex, L-DLPFC: left dorsolateral prefrontal cortex, R-DLPFC: right dorsolateral prefrontal cortex, L-MC: left motor cortex, R-MC: right motor cortex. (**C**) Experimental tasks and procedures. SC: single cognitive task, SW: single walking task, DT: dual-task.

**Figure 2 brainsci-13-01018-f002:**
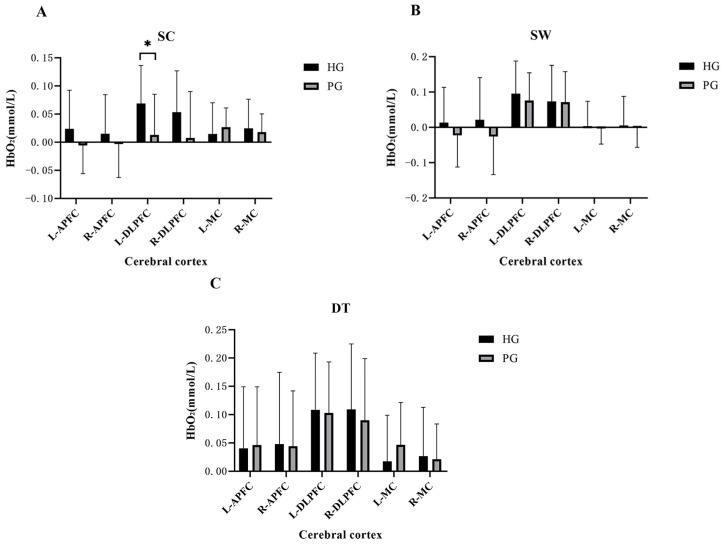
Cerebral cortex activation between the healthy and prefrail groups when performing (**A**) single cognitive task (SC), (**B**) single walking task (SW), and (**C**) dual-task (DT). L-APFC: left anterior prefrontal cortex, R-APFC: right anterior prefrontal cortex, L-DLPFC: left dorsolateral prefrontal cortex, R-DLPFC: right dorsolateral prefrontal cortex, L-MC: left motor cortex, R-MC: right motor cortex. * *p* < 0.05.

**Figure 3 brainsci-13-01018-f003:**
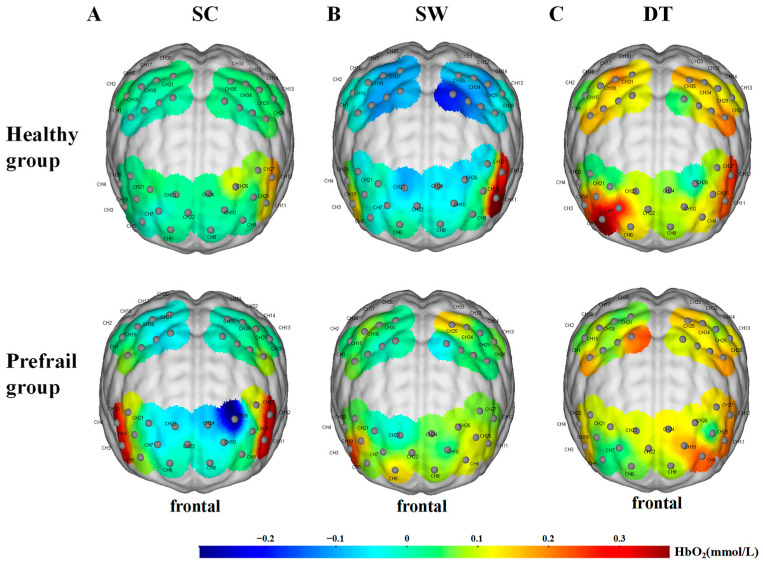
Activation of the cerebral cortex of a typical participant in the healthy and prefrail groups when performing (**A**) a single cognitive task (SC), (**B**) a single walking task (SW), and (**C**) a dual-task (DT). CH: channels. Red represents hyperactivation and blue represents hypoactivation.

**Figure 4 brainsci-13-01018-f004:**
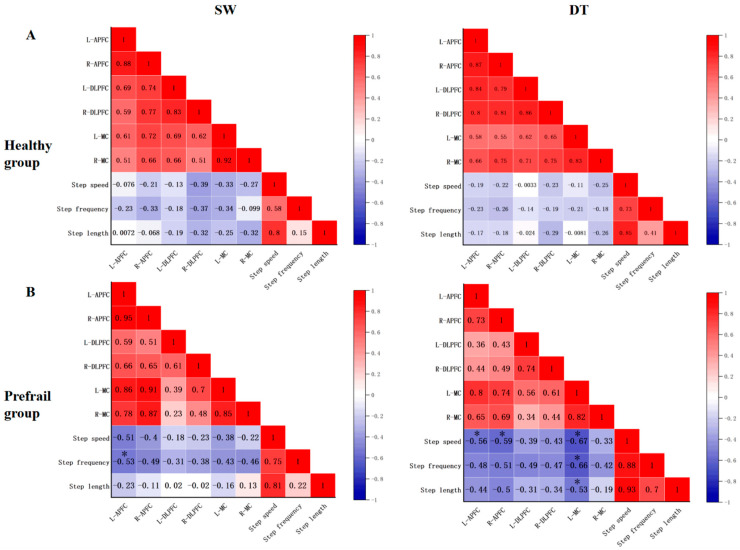
Heat map of the correlation between cerebral cortex activation and gait parameters during single walking task (SW) and dual-task (DT) in the (**A**) healthy group and (**B**) prefrail group. The number in the heat map is the correlation coefficient (*r*). L-APFC: left anterior prefrontal cortex, R-APFC: right anterior prefrontal cortex, L-DLPFC: left dorsolateral prefrontal cortex, R-DLPFC: right dorsolateral prefrontal cortex, L-MC: left motor cortex, R-MC: right motor cortex. * *p* < 0.05.

**Table 1 brainsci-13-01018-t001:** Frailty phenotype criteria.

Characteristics of Frailty	Specific Embodiment
1. Unintentional weight loss	At least 5% or more than 10 pounds of the previous year’s body weight
2. Exhaustion	Self-report in Epidemiological Studies Depression Scale
3. Low grip strength	Males:
	BMI ≤ 24 kg/m^2^, ≤29 kg; BMI 24.1–26 kg/m^2^, ≤30 kg;
	BMI 26.1–28 kg/m^2^, ≤30 kg; BMI > 28 kg/m^2^, ≤32 kg.
	Females:
	BMI ≤ 23 kg/m^2^, ≤17 kg; BMI 23.1–26 kg/m^2^, ≤17.3 kg;
	BMI 26.1–29 kg/m^2^, ≤18 kg; BMI > 29 kg/m^2^, ≤21 kg.
4. Slow pace	Males: height ≤ 173 cm, ≥7 s; height > 173 cm, ≥6 s
	Females: height ≤ 159 cm, ≥7 s; height > 159 cm, ≥6 s
5. Low physical activity	Males: <383 Kcals per week
	Females: <270 Kcals per week

BMI: body mass index.

**Table 2 brainsci-13-01018-t002:** Characteristics of the HG and PG (mean ± SD).

Variable	HG (N = 21)	PG (N = 15)	*p*-Value
Gender (female/all)	12/21	8/15	1.000
Age (years)	65.95 ± 3.81	67.93 ± 3.96	0.139
Body weight (kg)	62.67 ± 9.28	62.20 ± 6.12	0.866
Body height (cm)	160.95 ± 8.00	161.53 ± 6.88	0.822
BMI (kg/m^2^)	24.14 ± 2.67	23.89 ± 2.48	0.778
Years of education (years)	12.57 ± 2.56	11.60 ± 2.97	0.266
HGS (kg)	28.04 ± 6.14	25.15 ± 5.26	0.116
MoCA (maximum = 30)	27.14 ± 2.20	26.73 ± 1.39	0.529
CESD-10 (maximum = 30)	2.86 ± 2.65	3.27 ± 2.74	0.525
TUG (s)	9.25 ± 1.00	10.20 ± 1.03	0.010
IPAQ-SF (MET-min/week)	1242.90 ± 713.37	1026.93 ± 842.73	0.216
Frailty characteristics (number of participants)			
Unintentional body weight loss	/	2	/
Exhaustion	/	0	/
Low physical activity	/	6	/
Slow pace	/	6	/
Reduced grip strength	/	3	/

HG: healthy group, PG: prefrail group, BMI: body mass index, HGS: hand grip strength, MET: metabolic equivalent, SD: standard deviation, MoCA: Montreal Cognitive Assessment Scale, CESD-10: Short-Form of Center for Epidemiological Studies Depression Scale, TUG: Timed Up and Go test, IPAQ-SF: International Physical Activity Questionnaire-Short Form.

**Table 3 brainsci-13-01018-t003:** The 2-back task performance between the HG and PG when performing SC and DT (mean ± SD).

Variable	SC		*F*	*p*	DT		*F*	*p*
	HG (N = 21)	PG (N = 15)			HG (N = 21)	PG (N = 15)		
Accuracy (%)	64.88 ± 20.14	59.16 ± 16.18	0.826	0.261	57.27 ± 12.87	54.44 ± 15.61	0.354	0.556
Reaction time (ms)	1050.47 ± 144.48	1082.36 ± 133.69	0.453	0.505	1067.02 ± 124.45	1044.08 ± 159.72	0.235	0.987

SD: standard deviation, SC: single cognitive task, DT: dual-task, HG: healthy group, PG: prefrail group.

**Table 4 brainsci-13-01018-t004:** Gait performance between the HG and PG when performing SW and DT (mean ± SD).

Variable	SW		*F*	*p*	DT		*F*	*p*
	HG (N = 21)	PG (N = 15)			HG (N = 21)	PG (N = 15)		
Step speed (m/s)	1.20 ± 0.14	1.08 ± 0.13	6.607	0.015	1.13 ± 0.18	1.00 ± 0.20	4.033	0.053
Step frequency (steps/min)	113.38 ± 9.05	109.25 ± 8.38	1.929	0.174	110.96 ± 10.17	106.37 ± 12.93	1.422	0.241
Step length (m)	0.62 ± 0.07	0.59 ± 0.05	2.691	0.110	0.60 ± 0.07	0.56 ± 0.06	3.887	0.057
Step speed CV	4.64 ± 4.11	4.00 ± 2.56	0.284	0.642	4.34 ± 3.95	5.02 ± 5.49	0.188	0.665
Step frequency CV	3.11 ± 4.53	2.41 ± 1.67	0.329	0.936	2.70 ± 3.91	3.21 ± 5.54	0.106	0.785
Step length CV	2.27 ± 1.14	2.32 ± 1.34	0.018	0.860	1.92 ± 0.79	3.83 ± 5.24	2.748	0.026

SD: standard deviation, HG: healthy group, PG: prefrail group, SW: single walking task, DT: dual-task, CV: coefficient of variation.

## Data Availability

The data presented in this study are available on request from the corresponding author.

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
