# Peer review of "Cerebral Cortex Activation and Gait Performance between Healthy and Prefrail Older Adults during Cognitive and Walking Tasks"

_brainsci, 2023, doi:10.3390/brainsci13071018_

Round 1
Reviewer 1 Report
It cannot be ruled out that the measurement system used to extract cortical activity may influence the raw gait data obtained and complicate ecological plausibility.
It would be recommendable for future work to use data analysis methodologies of data signals that do not seek a traditional perspective in order to search for complex relationships in gait signals related to the search for biomarkers. (https://doi.org/10.3390/e21090868)
Author Response
Point 1: It cannot be ruled out that the measurement system used to extract cortical activity may influence the raw gait data obtained and complicate ecological plausibility.
Response 1: Thanks to reviewer for your careful and professional guidance. We think your concerns are reasonable. However, in recent years, due to the portability and movement tolerance of fNIRS equipment, it has been used in the study of neural mechanisms involving motor task paradigm (such as dual-task) [1-3]. Consistent with previous studies, our participants carried fNIRS devices on their backs, which had little effect on walking. Therefore, we believe that the impact of the fNIRS system on the raw gait data is relatively small.
Reference:
[1] Udina, C., Avtzi, S., Durduran, T., Holtzer, R., Rosso, A. L., Castellano-Tejedor, C., Perez, L. M., Soto-Bagaria, L., & Inzitari, M. (2020). Functional Near-Infrared Spectroscopy to Study Cerebral Hemodynamics in Older Adults During Cognitive and Motor Tasks: A Review. Frontiers in aging neuroscience, 11, 367. https://doi.org/10.3389/fnagi.2019.00367
[2] Gramigna, V., Pellegrino, G., Cerasa, A., Cutini, S., Vasta, R., Olivadese, G., Martino, I., & Quattrone, A. (2017). Near-Infrared Spectroscopy in Gait Disorders: Is It Time to Begin?.Neurorehabilitation and neural repair, 31(5), 402–412. https://doi.org/10.1177/1545968317693304
[3] Kahya, M., Moon, S., Ranchet, M., Vukas, R. R., Lyons, K. E., Pahwa, R., Akinwuntan, A., & Devos, H. (2019). Brain activity during dual task gait and balance in aging and age-related neurodegenerative conditions: A systematic review.Experimental gerontology, 128, 110756. https://doi.org/10.1016/j.exger.2019.110756
Point 2: It would be recommendable for future work to use data analysis methodologies of data signals that do not seek a traditional perspective in order to search for complex relationships in gait signals related to the search for biomarkers. (https://doi.org/10.3390/e21090868)
Response 2: We appreciate your professional and enlightening comments. We agree with your advice. In future research, we should take data analysis methods different from traditional data signals to further discover the complex relationships between gait signals and biomarkers. Just like the reference provided by the expert, it will also be novel and interesting to use permutation entropy and irreversibility in gait kinematic time series for future work about prefrail older adults.
Reviewer 2 Report
The paper by Fan et al. aims to explore cerebral cortex activation and gait performances between healthy and prefrail older adults during cognitive and walking tasks.
I have some suggestions to improve this work:
- Please state clearly the study design: was it a prospective, retrospective or a cross-sectional study ? How were patients (cases and controls) selected by the researchers (in particular, I am interested in the ratio of cases and controls)
- Please describe clearly the statistical power obtained by this small cohort with the adopted statistical methods, else report the low patients' number among study limitations (a small number with a reduced statistical power can translate into a type 0 or 1 error, thus the observed significance could be unreliable)
- Please use the word "p" instead of "P" when referring to the level of significance since this is the correct wording for this symbol
- Introduction and discussion are a little bit too long and should be reformatted to improve readability
- English form must be improved by a native English speaker. Several grammar errors and typos should be checked and corrected.
Author Response
Point 1: Please state clearly the study design: was it a prospective, retrospective or a cross-sectional study? How were patients (cases and controls) selected by the researchers (in particular, I am interested in the ratio of cases and controls)
Response 1: We feel great thanks for your professional review. We apologize for not clearly describing our study design in our paper. Our article is a cross-sectional study. We have stated our study design type in the abstract and methods section of the revised manuscript. Besides, the ratio of our sample size (cases and controls) is at least 1:1, and the specific sample size determination method is detailed in Response 2.
Point 2: Please describe clearly the statistical power obtained by this small cohort with the adopted statistical methods, else report the low patients' number among study limitations (a small number with a reduced statistical power can translate into a type 0 or 1 error, thus the observed significance could be unreliable)
Response 2: Thanks to reviewer for your careful guidance. We referred to the previous study and determined our sample size calculation method [1]. We used a moderate effect size of 0.4 (f), a power of 0.80, α-level at 0.05, and a correlation among the repeated measures of 0.4. The power analysis indicated that a minimum of 30 participants was required (at least 15 per group). Our sample size is 36 participants in total, which is consistent with the minimum sample size determined by a priori power analysis. The sample size calculation method has been clearly added to the revised manuscript. Please see page 3 of the revised manuscript, lines 103 - 105.
Reference:
[1] Goh, H. T., Pearce, M., & Vas, A. (2021). Task matters: an investigation on the effect of different secondary tasks on dual-task gait in older adults. BMC geriatrics, 21(1), 510. https://doi.org/10.1186/s12877-021-02464-8
Point 3: Please use the word "p" instead of "P" when referring to the level of significance since this is the correct wording for this symbol
Response 3: Thanks for your careful guidance. We feel sorry for our carelessness. We have changed all "P" to "p" in our revised manuscript.
Point 4: Introduction and discussion are a little bit too long and should be reformatted to improve readability
Response 4: Thank you for your constructive comments. Under the premise of complete background and discussion, we simplified the relevant statements in the introduction and discussion in order to improve readability.
Reviewer 3 Report
The authors describe differences in the neural activation between healthy and prefrail older adults using fNIRS when they performed dual-task and normal walking. The topic is suitable for the journal. My feedback that the authors have to reassess their work include the following:
1) Abstract:
Line 16: A short sentence can be improved to be “In this cross-sectional study, twenty-one healthy….. and fifteen prefrail older adults….”.
Line 20: Note it should be “healthy group”. Other places in the main text have the same issue.
Line 29: The involvement of the prefrontal to compensate for the impaired sensorimotor system … Does this occur in all task conditions or in DT only?
Negative correlation statement is difficult to comprehend at a glance. Replace the words ‘negative correlation’ with their interpretation, eg subjects with higher oxygenated area XX had lower step length etc. The statistics can still be included.
2) Participants
Line 120 – 127: I suggest the authors create a table for the different groups based on cutoff values.
Did the authors exclude older adults with walking stick who are still able to walk?
3) Clinical measures
Why did the authors ask for the TUG to walk comfortably instead of walking as fast (but safely) as possible?
Did the authors consider using TUG-only and TUG Dual-task for this instead? TUG Dual-task has been used previously, e.g. in Holfheinz et al,
https://pubmed.ncbi.nlm.nih.gov/20562166/
4) Procedures and Results
Did the authors look at the contribution of the jaw movement and movement artifacts on the NIRS signal?
I am puzzled that difference in activation only happened for SC although the cognitive measure has no difference. If the difference in clinical measure is only for TUG which is actually physical or kinematic parameter in nature. Does the prefrailty here specifically mean cognitively prefrail? How did the authors explain this?
Will the correlation of activity and parameters during other task conditions be meaningful if there is no difference in brain activation from the fNIRS data? This is to be clarified.
Acceptable
Author Response
Point 1: Line 16: A short sentence can be improved to be “In this cross-sectional study, twenty-one healthy….. and fifteen prefrail older adults….”.
Response 1: Thank you for your useful comments. We have simplified the sentence as suggested. Please see page 1 of the revised manuscript, line 16.
Point 2: Line 20: Note it should be “healthy group”. Other places in the main text have the same issue.
Response 2: Thanks to reviewer for your careful guidance. We feel sorry for our carelessness. In our resubmitted manuscript, the issues were revised. Thanks for your correction.
Point 3: Line 29: The involvement of the prefrontal to compensate for the impaired sensorimotor system … Does this occur in all task conditions or in DT only?
Response 3: Thank you for your careful review. According to our results, we found a negative correlation between prefrontal cortex activation and gait parameters in both SW and DT in PG (Figure 4). Therefore, this situation occurs in SW and DT.
Point 4: Negative correlation statement is difficult to comprehend at a glance. Replace the words ‘negative correlation’ with their interpretation, eg subjects with higher oxygenated area XX had lower step length etc. The statistics can still be included.
Response 4: We sincerely appreciate the valuable comments. We have changed the relevant statements according to your suggestion, as follows: “Participants of PG with higher oxygenated area in left anterior prefrontal cortex (L-APFC) had lower step frequency during SW (r = −0.533, p = 0.041). So did the following indicators of PG during DT: L-APFC and step speed (r = −0.557, p = 0.031); right anterior prefrontal cortex and step speed (r = −0.610, p = 0.016); left motor cortex and step speed (r = −0.674, p = 0.006); step frequency (r = −0.656, p = 0.008); and step length (r = −0.535, p = 0.040).” Please see page 1 of the revised manuscript, lines 24 - 29.
Point 5: Line 120 – 127: I suggest the authors create a table for the different groups based on cutoff values.
Response 5: Thanks very much for your valuable comments. In order to clarify the classification of frailty, we have added a table (Table 1) for the different groups based on cutoff values. Please see page 3 of the revised manuscript, lines 118 - 119.
Point 6: Did the authors exclude older adults with walking stick who are still able to walk?
Response 6: Thank you for your careful review. In our study, we excluded the older adults who used walking sticks. The exclusion criteria in our article have already mentioned this point (i.e. limb movement disorder and obstacles to walking).
Point 7: Why did the authors ask for the TUG to walk comfortably instead of walking as fast (but safely) as possible?
Response 7: We appreciate your professional question. We let the participants walk comfortably to simulate the real walking state of the participants in daily life, so as to reflect the real walking level of the participants.
Point 8: Did the authors consider using TUG-only and TUG Dual-task for this instead? TUG Dual-task has been used previously, e.g. in Holfheinz et al,
https://pubmed.ncbi.nlm.nih.gov/20562166/
Response 8: Thank you for your professional and enlightening comments. TUG dual-task was not considered in our existing studies, partly because the time of the TUG test was too short, and too short time may not be suitable for a hemodynamic response of fNIRS (one consideration when processing fNIRS signals is the time-lag of 4-7 s between cortical activity and hemodynamic response) [1,2]. However, the combination of fNIRS and TUG dual-task may be an interesting discovery, which is worthy of further study.
Reference:
[1] Tong, Y., & Frederick, B. D. (2010). Time lag dependent multimodal processing of concurrent fMRI and near-infrared spectroscopy (NIRS) data suggests a global circulatory origin for low-frequency oscillation signals in human brain.NeuroImage, 53(2), 553–564.https://doi.org/10.1016/j.neuroimage.2010.06.049
[2] Cui, X., Bray, S., & Reiss, A. L. (2010). Speeded near infrared spectroscopy (NIRS) response detection.PloS one, 5(11), e15474. https://doi.org/10.1371/journal.pone.0015474
Point 9: Did the authors look at the contribution of the jaw movement and movement artifacts on the NIRS signal?
Response 9: We appreciate your professional question. We considered the impact of jaw movement and motion artifacts on fNIRS signal. At present, fNIRS optical methods are indeed affected by large amplitude motion. However, the range of motion in the walking task in our experimental paradigm was relatively small, and the participants were instructed to minimize movements unrelated to the execution of the task (e.g. avoiding excessive head flexion /extension and talking) [1,3]. Besides, our fNIRS light source probe fits snugly to the participants’ scalps, reducing motion artifacts for maximal light coupling efficiency to tissues [2]. Moreover, compared with other neuroimaging techniques (e.g. EEG and fMRI), fNIRS is both portable and less sensitive to movement artifacts [3]. Lastly, the introduction of motion artifact removal techniques will add to the device’s robustness and stability even when the users are involved in significant physical activities [1]. Lastly, to further improve the signal-to-noise ratio of HbO2, the spline interpolation methods were used to eliminate the motion artifacts and noises. All the above methods have been used in our study.
Reference:
[1] Huang, R., Hong, K. S., Yang, D., & Huang, G. (2022). Motion artifacts removal and evaluation techniques for functional near-infrared spectroscopy signals: A review. Frontiers in neuroscience, 16, 878750. https://doi.org/10.3389/fnins.2022.878750
[2] Park, E., Kang, M. J., Lee, A., Chang, W. H., Shin, Y. I., & Kim, Y. H. (2017). Real-time measurement of cerebral blood flow during and after repetitive transcranial magnetic stimulation: A near-infrared spectroscopy study. Neuroscience letters, 653, 78–83. https://doi.org/10.1016/j.neulet.2017.05.039
[3] Menant, J. C., Maidan, I., Alcock, L., Al-Yahya, E., Cerasa, A., Clark, D. J., de Bruin, E. D., Fraser, S., Gramigna, V., Hamacher, D., Herold, F., Holtzer, R., Izzetoglu, M., Lim, S., Pantall, A., Pelicioni, P., Peters, S., Rosso, A. L., St George, R., Stuart, S., … Mirelman, A. (2020). A consensus guide to using functional near-infrared spectroscopy in posture and gait research. Gait & posture, 82, 254–265. https://doi.org/10.1016/j.gaitpost.2020.09.012
Point 10: I am puzzled that difference in activation only happened for SC although the cognitive measure has no difference. If the difference in clinical measure is only for TUG which is actually physical or kinematic parameter in nature. Does the prefrailty here specifically mean cognitively prefrail? How did the authors explain this?
Response 10: Thank you for your careful and professional review. The concept of cognitive frailty was introduced to encompass a broader definition of frailty, where an older adult exhibits the simultaneous presence of physical frailty and cognitive impairment in the absence of dementia [1].
Our results showed that although the 2-back performance and MoCA scores of the two groups were normal, there were differences in cerebral cortex activation (L-DLPFC) between the two groups when performing SC. Therefore, we believe that the performance in SC may be an early indicator of cognitive frailty in prefrail older adults. Indicating that the change of HBO2 in the cortex can more sensitively detect the cognitive decline of prefrail older adults. Besides, our TUG test results also provide evidence for their physical frailty. Our findings are consistent with the concept of cognitive frailty. Therefore, it can be considered that the prefrail older adults in our study not only have physical frailty, but also have the possibility of cognitive frailty. This result further proves the application prospect of fNIRS in the discovery of frailty biomarkers.
Reference:
[1] Kelaiditi, E., Cesari, M., Canevelli, M., van Kan, G. A., Ousset, P. J., Gillette-Guyonnet, S., Ritz, P., Duveau, F., Soto, M. E., Provencher, V., Nourhashemi, F., Salvà, A., Robert, P., Andrieu, S., Rolland, Y., Touchon, J., Fitten, J. L., Vellas, B., & IANA/IAGG (2013). Cognitive frailty: rational and definition from an (I.A.N.A./I.A.G.G.) international consensus group. The journal of nutrition, health & aging, 17(9), 726–734. https://doi.org/10.1007/s12603-013-0367-2
Point 11: Will the correlation of activity and parameters during other task conditions be meaningful if there is no difference in brain activation from the fNIRS data? This is to be clarified.
Response 11: Thank you for your careful and professional review. Although we only found significance between the two groups in SC, this did not affect our intra-group correlation analysis of cerebral cortex activation and gait parameters. Because previous studies have indicated that the cortical regions are related to human motor control (such as prefrontal cortex and motor cortex) [1]. In our study, a significant moderate negative correlation was noted between gait and cerebral cortex activation, including the L-APFC, R-APFC, and L-MC, in the PG although non-existent in the HG. Our study is the confirmation of previous studies, which further proves that the neuro compensation effect exists in prefrail older adults, that is, the activation of prefrontal cortex is to compensate for the decline of gait function in prefrail older adults [2].
Reference:
[1] Hamacher, D., Herold, F., Wiegel, P., Hamacher, D., & Schega, L. (2015). Brain activity during walking: A systematic review. Neuroscience and biobehavioral reviews, 57, 310– https://doi.org/10.1016/j.neubiorev.2015.08.002
[2] Poole, V. N., Wooten, T., Iloputaife, I., Milberg, W., Esterman, M., & Lipsitz, L. A. (2018). Compromised prefrontal structure and function are associated with slower walking in older adults. NeuroImage. Clinical, 20, 620–626. https://doi.org/10.1016/j.nicl.2018.08.017
Reviewer 4 Report
This study investigated the relationship between gait parameters and the neural activity of walking in pre-frail subjects. This is an important finding in clarifying gait dynamics in pre-frail subjects. However, there was a point of concern regarding the method of measuring and analyzing brain activity.
Page 3, line 106
Does it meet the required sample size? What is the appropriate sample size needed for this study?
Page 3, line 139
Right turn? Left turn? Additionally, is the direction of the turn standardized?
Page 4, line 179
Please cite previous study.
Page 4, line 186
Spatial coordinates shift somewhat from person to person. How did you standardize the coordinates of 14 or 21 subject? Are brain volumes If they are different, then the standardization of MNI coordinates for pre-frail only should be group specific.
Page 5, line 207
The study was a repeated block design; Oxy-Hb increased over time, with a higher slope. Do you perform baseline correction during analysis?
Page 5, 195
Why did you measure all conditions in one block? Could they be affected by the previous task? In particular, SW after DT could be affected by DT because the task is easier. I thought it would be better to measure separately and take a break to minimize the impact of the assignment.
Page6, 229
Did you analyze 40m for CV? Clearly.
Page 7, line 276
Why did you apply the test to each area?
Wouldn't the spatial resolution brain be very low?
In addition, if you look at mapping, it is possible that certain channels are pulling values. If that is the case, it would be better to test by channel and then correct for whole brain, which would clearly reveal the task-specific relevant brain regions.
Page 13, line 417
If the pre-frail neurology and gait dynamics were more clearly defined, the frail subject could have been added.
-
Author Response
Point 1: Page 3, line 106
Does it meet the required sample size? What is the appropriate sample size needed for this study?
Response 1: We feel great thanks for your professional review. We referred to previous study and determined our sample size calculation method [1]. We used a moderate effect size of 0.4 (f), a power of 0.80, α-level at 0.05, and a correlation among the repeated measures of 0.4. The power analysis indicated that a minimum of 30 participants was required (at least 15 per group). Our sample size is 36 participants in total, which is consistent with the minimum sample size determined by a priori power analysis. The sample size calculation method has been clearly added to the revised manuscript. Please see page 3 of the revised manuscript, lines 103 - 105.
Reference:
[1] Goh, H. T., Pearce, M., & Vas, A. (2021). Task matters: an investigation on the effect of different secondary tasks on dual-task gait in older adults. BMC geriatrics, 21(1), 510. https://doi.org/10.1186/s12877-021-02464-8
Point 2: Page 3, line 139
Right turn? Left turn? Additionally, is the direction of the turn standardized?
Response 2: Thank you for your careful review. Our participants are right-handed. Therefore, we uniformly guide participants to turn right during the TUG test in our study.
Point 3: Page 4, line 179
Please cite previous study.
Response 3: We sincerely appreciate the valuable comments. We have added a reference about the device used in our study, which was also used in previous studies. (DOI: 10.1016/j.neuroimage.2022.119028)
Point 4: Page 4, line 186
Spatial coordinates shift somewhat from person to person. How did you standardize the coordinates of 14 or 21 subject? Are brain volumes If they are different, then the standardization of MNI coordinates for pre-frail only should be group specific.
Response 4: Thank you for your careful and professional review. We acknowledge that there are certain changes in spatial coordinates between people. However, we have standardized the measurement of fNIRS. The experiment uses 13 sources and 15 detectors to form 35 channels, and the average distance between the source and the detector is 2.7 cm, with reference to the international 10/20 system for positioning. The acquired coordinates were then transformed into Montreal Neurological Institute coordinates and further projected to the Montreal Neurological Institute standard brain template using a spatial registration approach in NirSpace. The standardization of head cap positioning above reduces measurement differences within and between groups.
Point 5: Page 5, line 207
The study was a repeated block design; Oxy-Hb increased over time, with a higher slope. Do you perform baseline correction during analysis?
Response 5: We sincerely appreciate your professional review. Baseline correction is included in our fNIRS data processing. All Oxy-Hb data were performed baseline correction during analysis.
Point 6: Page 5, 195
Why did you measure all conditions in one block? Could they be affected by the previous task? In particular, SW after DT could be affected by DT because the task is easier. I thought it would be better to measure separately and take a break to minimize the impact of the assignment.
Response 6: Thank you for your careful review. We referred to previous studies and measured all conditions in one block [1,2]. In order to effectively control the fatigue effect of the participants under various conditions. The stimulus block was presented according to the A-B-B-A design (each experimental condition was managed in a block of 30 s; subsequently, the participants were instructed to rest for 30 s) (Figure 1C). The reliability and validity of this paradigm have been well-established by previous study [1]. Indeed, we agree with you, it may be more appropriate to take the method of separate measurement to minimize the impact of the assignment in future research.
Reference:
[1] Doi, T., Makizako, H., Shimada, H., Park, H., Tsutsumimoto, K., Uemura, K., & Suzuki, T. (2013). Brain activation during dual-task walking and executive function among older adults with mild cognitive impairment: a fNIRS study. Aging clinical and experimental research, 25(5), 539– https://doi.org/10.1007/s40520-013-0119-5
[2] Talamonti, D., Vincent, T., Fraser, S., Nigam, A., Lesage, F., & Bherer, L. (2021). The Benefits of Physical Activity in Individuals with Cardiovascular Risk Factors: A Longitudinal Investigation Using fNIRS and Dual-Task Walking.Journal of clinical medicine, 10(4), 579. https://doi.org/10.3390/jcm10040579
Point 7: Page6, 229
Did you analyze 40m for CV? Clearly.
Response 7: Thank you for your careful review. The 40-meter corridor is designed to ensure that the participants keep straight without turning during the 30-second walking time. We analyzed the coefficient of variation of all gait data within 30 seconds in each block.
Point 8: Page 7, line 276
Why did you apply the test to each area?
Wouldn't the spatial resolution brain be very low?
In addition, if you look at mapping, it is possible that certain channels are pulling values. If that is the case, it would be better to test by channel and then correct for whole brain, which would clearly reveal the task-specific relevant brain regions.
Response 8: Thanks for your careful review. A portable multichannel fNIRS device was used in our study. According to previous studies, we selected brain regions of interest (ROI), which are related to our cognitive motor (dual-task) paradigm [1,2]. These ROI are composed of multiple channels, so we tested each area.
The major drawback of fNIRS is its lower spatial resolution (few centimeters under the skull) and its lack of sensitivity to subcortical regions [3,4]. However, this might be considered a minor limitation, as there is a large body of evidence suggesting that cortical mechanisms take place in walking and cognitive task [1,3].
We agree with your suggestion very much. In future research, we can conduct channel analysis and then correct the whole brain, which would clearly reveal the brain regions related to specific tasks.
Reference:
[1] Udina, C., Avtzi, S., Durduran, T., Holtzer, R., Rosso, A. L., Castellano-Tejedor, C., Perez, L. M., Soto-Bagaria, L., & Inzitari, M. (2020). Functional Near-Infrared Spectroscopy to Study Cerebral Hemodynamics in Older Adults During Cognitive and Motor Tasks: A Review. Frontiers in aging neuroscience, 11, 367. https://doi.org/10.3389/fnagi.2019.00367
[2] Lin, C. C., Bair, W. N., & Willson, J. (2022). Age differences in brain activity in dorsolateral prefrontal cortex and supplementary motor areas during three different walking speed tasks. Human movement science, 85, 102982. https://doi.org/10.1016/j.humov.2022.102982
[3] Gramigna, V., Pellegrino, G., Cerasa, A., Cutini, S., Vasta, R., Olivadese, G., Martino, I., & Quattrone, A. (2017). Near-Infrared Spectroscopy in Gait Disorders: Is It Time to Begin?. Neurorehabilitation and neural repair, 31(5), 402–412. https://doi.org/10.1177/1545968317693304
[4] Kahya, M., Moon, S., Ranchet, M., Vukas, R. R., Lyons, K. E., Pahwa, R., Akinwuntan, A., & Devos, H. (2019). Brain activity during dual task gait and balance in aging and age-related neurodegenerative conditions: A systematic review. Experimental gerontology, 128, 110756. https://doi.org/10.1016/j.exger.2019.110756
Point 9: Page 13, line 417
If the pre-frail neurology and gait dynamics were more clearly defined, the frail subject could have been added.
Response 9: Thank you for your enlightening comments. In our study, we aimed to explore the central mechanisms of cognitive and gait deficits associated with health and pre-frailty. We found the neurologic and gait deficits of the prefrail older adults. In our current study, the frail older adults were not included. Therefore, based on the existing work, we will continue to carry out research on frail older adults in the future.
Round 2
Reviewer 3 Report
The authors have adequately addressed the concerns.
Reviewer 4 Report
Thank you for variable comments.
-